

# The contribution of Earth observation technologies to the reporting obligations of the Habitats Directive and Natura 2000 network in a protected wetland

Adrián Regos[1,2] and Jesús Domínguez[1]

[1] Departamento de Zooloxía, Xenética e Antropoloxía Física, Universidade de Santiago de Compostela, Santiago de Compostela, Spain

[2] Predictive Ecology Group, Centro de Investigacão em Biodiversidade e Recursos Genéticos da Universidade do Porto, CIBIO/InBIO, Vairão, Portugal

## ABSTRACT

**Background**. Wetlands are highly productive systems that supply a host of ecosystem services and benefits. Nonetheless, wetlands have been drained and filled to provide sites for building houses and roads and for establishing farmland, with an estimated worldwide loss of 64–71% of wetland systems since 1900. In Europe, the Natura 2000 network is the cornerstone of current conservation strategies. Every six years, Member States must report on implementation of the European Habitats Directive. The present study aims to illustrate how Earth observation (EO) technologies can contribute to the reporting obligations of the Habitats Directive and Natura 2000 network in relation to wetland ecosystems.

**Methods**. We analysed the habitat changes that occurred in a protected wetland (in NW Spain), 13 years after its designation as Natura 2000 site (i.e., between 2003 and 2016). For this purpose, we analysed optical multispectral bands and water-related and vegetation indices derived from data acquired by Landsat 7 TM, ETM+ and Landsat 8 OLI sensors. To quantify the uncertainty arising from the algorithm used in the classification procedure and its impact on the change analysis, we compared the habitat change estimates obtained using 10 different classification algorithms and two ensemble classification approaches (majority and weighted vote).

**Results**. The habitat maps derived from the ensemble approaches showed an overall accuracy of 94% for the 2003 data (Kappa index of 0.93) and of 95% for the 2016 data (Kappa index of 0.94). The change analysis revealed important temporal dynamics between 2003 and 2016 for the habitat classes identified in the study area. However, these changes depended on the classification algorithm used. The habitat maps obtained from the two ensemble classification approaches showed a reduction in habitat classes dominated by salt marshes and meadows (24.6–26.5%), natural and semi-natural grasslands (25.9–26.5%) or sand dunes (20.7–20.9%) and an increase in forest (31–34%) and reed bed (60.7–67.2%) in the study area.

**Discussion**. This study illustrates how EO–based approaches might be particularly useful to help (1) managers to reach decisions in relation to conservation, (2) Member States to comply with the requirements of the European Habitats Directive (92/43/EEC), and (3) the European Commission to monitor the conservation status of the natural habitat types of community interest listed in Annex I of the Directive.

Corresponding author
Adrián Regos, adrian.regos@usc.es

Nonetheless, the uncertainty arising from the large variety of classification methods used may prevent local managers from basing their decisions on EO data. Our results shed light on how different classification algorithms may provide very different quantitative estimates, especially for water-dependent habitats. Our findings confirm the need to account for this uncertainty by applying ensemble classification approaches, which improve the accuracy and stability of remote sensing image classification.

## INTRODUCTION

Wetlands are highly productive systems that provide a host of ecosystem services and benefits, including local climate regulation, erosion control, recreational fishing, flood control and long-term supply of good quality ground water, storage of pollutants, rare species habitat, and cultural heritage and educational value (*De Groot et al., 2006*; *Horwitz & Finlayson, 2011*). Nonetheless, wetlands have been perceived as a source of vectors of waterborne infectious diseases, and historically considered worthless and an impediment to development. Consequently, wetlands have been drained and filled to provide sites for building houses and roads or for establishing farmland, with an estimated worldwide loss of 64–71% of wetland systems since 1900 (*Davidson, 2014*).

Protection for wetlands can come in many forms, ranging from local practices and national legislation to international recognition through inscription on the Ramsar List and/or the World Heritage List (*Thorsell, Levy & Sigaty, 1997*). In Europe, the Natura 2000 network is the cornerstone of current environmental conservation strategies. This network includes Special Protection Areas for wild birds (SPAs), designated by the Member States under the Birds Directive (2009/147/EC) with the aim of conserving the habitats of particularly threatened species and migratory species. It also includes Special Areas of Conservation (SACs), designated for other taxa and habitats under the Habitats Directive (92/43/ EEC). Every six years, Member States must report on implementation of the measures taken under these European Directives. This report must include information on the conservation measures concerning the natural habitat types listed in Annex I of the Habitats Directive (Art. 6), as well as evaluation of the impacts and surveillance (Art. 2) of those measures in relation to their conservation status, with particular regard to priority natural habitat types and priority species.

Earth observation (EO) technologies have made significant contributions to nature conservation in the last few decades (*Muchoney, 2008*; *O'Connor et al., 2015* and reference therein). Increasingly large amounts of geospatial information are being provided by satellite and aerial image processing and analysis—also known as remote sensing (RS)—which has enormous potential for conservation applications (*Leyequien et al., 2007*; *Alcaraz-Segura et al., 2009*; *Petrou, Manakos & Stathaki, 2015*; *Skidmore et al., 2015*;

*Adamo et al., 2016*, among others). Access to EO data has improved greatly in recent years, and many aerial and satellite data are now freely available (*Turner et al., 2015*).

Despite the above-mentioned progress, the lack of a single, unifying habitat feature as well as the highly dynamic nature of wetlands (which may lead to highly variable spectral signatures) and their steep environmental gradients (which often produce narrow ecotone areas) may constrain and overwhelm the capacity of current remote sensors (*Gallant, 2015*). Recent advances in computing and the development of image classification techniques have made RS-based land-cover mapping easier, faster and more widely available for use in both conservation and applied ecology (*Khatami, Mountrakis & Stehman, 2016*). Faced with this wide range of techniques, many researchers have focused on comparing the image classification performance of land-cover mapping or other applications (e.g., *Hubert-Moy et al., 2001*; *Cracknell & Reading, 2014*; *Regos et al., 2015*). One effective solution for dealing with the uncertainty arising from the use of a wide range of techniques is to generate a classification ensemble by combining some individual classifiers. This is referred to as a multiple classification system or ensemble classification approach (for a review, see *Du et al., 2012*). The ensemble classification approach, recently applied by the remote sensing community, is viewed as an effective way of improving the classification performance of remotely sensed imagery (*Briem, Benediktsson & Sveinsson, 2002*; *Lu & Weng, 2007*).

The main goal of the present work is to illustrate how EO technologies may contribute to the reporting obligations of the Habitats Directive and Natura 2000 network regarding wetland ecosystems. We analysed the habitat changes that have taken place in a protected wetland (in NW Spain), 13 years after its designation as Natura 2000 site (2003–2016). For this purpose, we analysed optical multispectral bands and water-related and vegetation indices derived from data captured by Landsat 7 TM, ETM+ and Landsat 8 OLI sensors. To quantify the uncertainty arising from the algorithm used in the classification procedure and its impact on the change analysis, we compared the habitat change estimates obtained using 10 different classification algorithms and two ensemble classification approaches.

## MATERIAL & METHODS

### Study site

The study area is a coastal wetland included in the Natura 2000 network in 2003 and designated as Special Area of Conservation (SAC) and Special Protection Area (SPA) for wild birds. The site covers an area of 984 ha, corresponding to the boundaries of the "Dunas de Corrubedo e lagoas de Carregal e Vixán" Natural Park (Fig. 1). The international importance of the wetland was recognised when it was designated a Ramsar site, in 1993.

This wetland includes one of the largest dune systems in the NW Iberian Peninsula, with extensive stretches of sand (Ladeira, Ferreira and Vilar beaches) flanked by large dune and coastal lagoon ecosystems (Lagunas de Carregal and Vixán), together with an adjacent dune system, and an embryonic shifting dune (1-km long, 200–250 m wide and 12–15 m high) (*Vázquez-Paz & Pérez-Alberti, 2002*). The dune system, comprising a sandy barrier, has favoured the creation of an interior sedimentary area composed of fixed dunes ('grey dunes'), marshes, sandy and muddy intertidal zones, as well as two coastal lagoons

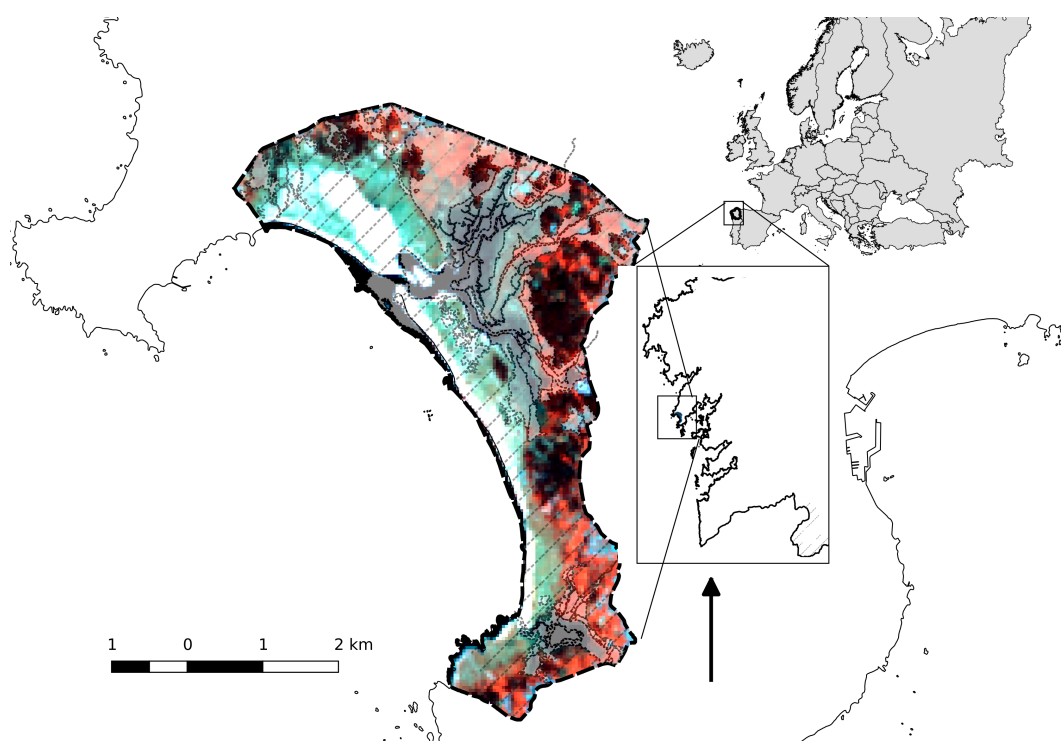

**Figure 1 Location of the study area and protected-area systems.** Ramsar wetland (dashed-dotted line), Natural park and SAC (black dashed line) and SPA (filling lines).

with very different aquatic characteristics: (1) the Carregal lagoon covers an irregular, delimited space between the marsh and the dune system. The area adjacent to the coastline corresponds morphologically to an estuarine channel covered by sandy deposits where flooding depends on the tidal cycle (Fig. 1); (2) the Vixán lagoon, located in the area distal to the coastline, has a dense reed bed (*Phragmites australis*) that occupies most of the eulittoral and supralittoral environments (Fig. 1). In the area adjacent to the coast, the reed bed is replaced by bulrushes (*Typha latipholia*) and, to a lesser extent, by wet grasslands. The drainage channel zigzags through the dune system until reaching the beach (*Ramil-Rego, 2007*).

## Pre-processing EO data

We used satellite remote sensing imagery to map and monitor the habitat changes that have taken place between 2003 and 2016. We analysed optical multispectral bands (Path/Row: 205/30) derived from four cloud-free images acquired by NASA's Landsat missions on 20 March (Landsat 7 ETM+) and 6 October 2003 (Landsat 5 TM) and on 2 May and 23 September 2016 (Landsat 8 OLI) (detailed information available for each band is available at: http://landsat.usgs.gov/band_designations_landsat_satellites.php). Landsat scenes captured in spring and autumn (e.g., in May and September) were analysed to take into account seasonal differences in vegetation phenology (e.g., common reed grass). The images are all available free of charge from the US Geological Survey (USGS) Centre for

Earth Resources Observation and Science (EROS) and were obtained by direct download from the GloVis facility (http://glovis.usgs.gov).

All downloaded images were L1T (a processing level that includes a geometric correction performed with ground control points and the use of a digital elevation model) and projected in the UTM coordinate system (WGS 84 datum, UTM projection, Zone 29 North). Digital numbers (DNs) were converted to top-of-atmosphere radiance and physically meaningful units by radiometric calibration and application of sensor- and band-specific calibration parameters. The classification process was based on the radiometric information obtained from reflective bands and two multispectral indices for each image: (1) the Normalized Difference Vegetation Index (NDVI; *Rouse et al., 1974*) and (2) the Normalized Difference Water Index (NDWI; *Gao, 1996*). This procedure enhanced the spectral separability of vegetation associated with aquatic and halophilic environments.

## Classification procedure

Supervised classification of the remotely-sensed data was carried out using the following 10 classification algorithms available in the R-based package *Caret* and implemented in the RStoolbox package, version 0.1.5 (*Kuhn, 2017*; *Leutner & Horning, 2017*): amdai (Adaptive Mixture Discriminant Analysis), avNNet (Model Averaged Neural Network), gbm (Stochastic Gradient Boosting), knn (k-Nearest Neighbours), mda (Mixture Discriminant Analysis), pls (Partial Least Squares), rf (Random Forest), svmPoly (Support Vector Machines with Polynomial Kernel), svmLinear (Support Vector Machines with Linear Kernel) and svmRadial (Support Vector Machines with Radial Basis Function Kernel). In addition, two ensemble procedures were performed: (1) a simple voting system ('Ens_SV'; the so-called 'majority voting' and 'select all majority' system, sensu *Bauer et al., 1999*), considering each habitat map as an equally weighted vote; and (2) a weighted voting approach ('Ens_WV'), using overall accuracy obtained by individual classifiers as weights (*Du et al., 2012*).

Eight habitat classes, defined as areas with common ecological and biophysical characteristics and, therefore, with a homogeneous spectral signature, were identified in the study area. For these habitat classes, we adopted the terminology used in the Annex I of the Habitats Directive. These eight habitat classes correspond with 23 specific habitats listed in this Annex I in our study area (Table 1). The study area is very well described, and the whole list of habitats is already defined in previous reports (see e.g., *Ramil-Rego et al., 2008*). Training and validation areas for each habitat class were established by on-screen digitizing in QGIS software, and consisted of a set of pixels identified over well-known homogeneous areas in each Landsat image, thus providing a reference spectral signature for each class. In particular, we applied a stratified random design as sampling strategy, with a total of about 259–346 training and validation areas proportionally distributed throughout the entire study area for each year (Table 2; Dataset S1). Specifically, for 2003 we used different Red-Green-Blue (RGB) composites from the Landsat bands and digital orthophotos in natural colours at a scale of 1:18,000 obtained from the *Plan Nacional de Ortofotografía Aérea* (PNOA) for 2004, while for 2016 we used digital orthophotos from 2014.

**Table 1** **List of broad habitat classes used in the change analysis and their correspondence with the natural habitats (and codes) listed in the Annex I of the Habitats Directive.** Asterisk indicates habitats with highest priority for conservation according to the Habitats Directive.

| Habitat class | Natural habitats listed in the Annex I of the Habitat Directive |
|---|---|
| Sand dunes | 1110 Sandbanks which are slightly covered by sea water all the time.<br>1140 Mud flats and sandflats not covered by sea water at low tide.<br>1210 Annual vegetation of drift lines.<br>2110 Embryonic shifting dunes.<br>2120 Shifting dunes along the shoreline with *Ammophila arenaria* ('white dunes'). |
| Tidal areas | 1130 Estuaries.<br>1150* Coastal lagoons.<br>1160 Large shallow inlets and bays.<br>1170 Reefs. |
| Forest | – |
| Reedbed | – |
| Sea dunes of Atlantic coast | 2130* Fixed coastal dunes with herbaceous vegetation ('grey dunes')<br>2150* Atlantic decalcified fixed dunes (*Calluno-Ulicetea*).<br>2190 Humid dune slacks.<br>2230 *Malcolmietalia* dune grasslands.<br>2260 *Cisto-Lavenduletalia* dune sclerophyllous scrubs. |
| Natural and semi-natural grasslands | 6220* Pseudo-steppe with grasses and annuals of the *Thero-Brachypodietea*<br>6410 Molinia meadows on calcareous, peaty or clayey-silt-laden soils (*Molinion caeruleae*).<br>6420 Mediterranean tall humid grasslands of the *Molinio-Holoschoenion*.<br>6430 Hydrophilous tall herb fringe communities of plains and of the montane to alpine levels.<br>6510 Lowland hay meadows (*Alopecurus pratensis*, *Sanguisorba officinalis*) |
| Salt marshes and meadows | 1310 *Salicornia* and other annuals colonizing mud and sand.<br>1330 Atlantic salt meadows (*Glauco-Puccinellietalia maritimae*).<br>1420 Mediterranean and thermo-Atlantic halophilous scrubs (*Sarcocornetea fruticosi*). |
| Burned areas | – |

**Table 2** **Total number of training and validation areas considered in the supervised classification for each habitat class and year.**

| Habitat class | Training | Validation | Training | Validation |
|---|---|---|---|---|
| | **2003** | | **2016** | |
| Sand dunes | 38 | 30 | 35 | 29 |
| Tidal areas | 48 | 29 | 49 | 31 |
| Forest | 64 | 61 | 76 | 43 |
| Reedbed | 21 | 22 | 20 | 23 |
| Sea dunes of Atlantic coast | 46 | 29 | 56 | 29 |
| Natural and semi-natural grasslands | 31 | 55 | 44 | 39 |
| Salt marshes and meadows | 34 | 33 | 33 | 26 |
| Burned areas | 0 | 0 | 39 | 23 |
| TOTAL | 282 | 259 | 346 | 243 |

## Validation procedure

The accuracy of habitat maps was assessed from confusion matrices based on the number of pixels correctly (and incorrectly) classified per class, and by comparing the results obtained from different classification algorithms. The main quality parameters were the overall accuracy (%), the producer's and user's accuracies, and the Kappa index of agreement.

We used McNemar's tests to evaluate statistical significance of the difference in accuracy between each pair of algorithms. This is a non-parametric test that is based on confusion matrices collapsed to two by two contingency tables (*Foody, 2004*; *De Leeuw et al., 2006*). *P*-values from McNemar's tests were represented with heatmaps to help visualizing statistical significance of the difference between all possible comparisons. These *p*-values were used to support the selection of algorithms for the subsequent ensemble procedures. Thereby, classification algorithms with statistically lower accuracies were not included in the ensemble procedures (McNemar's tests, $p < 0.05$).

Data importation, pre-processing, spectral indices, image classification and graphical display were performed using the toolset available in RStoolbox package, version 0.1.5 (*Wegmann, Leutner & Dech, 2016*; *Leutner & Horning, 2017*) (see http://rpubs.com/ARegos/359655 for R code and formatted outputs).

### Change analysis

We quantified the spatial extent (in ha) of each habitat class per year (2003 and 2016) from each classification algorithm and ensemble approach. Boxplots were constructed using the R package *ggplot2* (*Wickham, 2009*). The contribution of each habitat class to the habitat change (i.e., conversion from one habitat class to another) was showed through a transition matrix obtained by cross-tabulation of the habitat maps derived from the two ensemble classification approaches. Transition matrices were computed with the R package *lulcc* v.1.0.2 (*Moulds, 2017*) (see http://rpubs.com/ARegos/359655 for R code and formatted outputs).

## RESULTS

### Accuracy assessment

The habitat maps with the highest accuracy (up to 95%) in 2003 were obtained using support vector machines and discriminant analysis, with the 'amdai' classifier providing slightly better results (Fig. 2). For 2016, the highest accuracy was obtained by applying support vector machines with linear kernel (Fig. 2). However, McNemar's test did not show statistical significance of the difference in accuracy between individual classification algorithms ($p > 0.05$; Fig. 3), except for 'pls', 'avNNet' ($p < 0.01$; Fig. 3), 'gbm and 'svmRadial' ($p < 0.05$; Fig. 3). These algorithms showed limitations for specific habitat classes that have led to under- and overestimations of their extent (Fig. 4). For instance, 'pls' showed very low user's accuracies for the thematic class 'forest', while 'svmRadial' markedly overestimated the habitat class 'tidal area' (see low user's accuracy and high producer's accuracy values, i.e., low omission errors and high commission errors in Fig. 4). Thereby, 'pls' and 'avNNet' for both years and 'gbm' and 'svmRadial' for year 2016 were finally not considered during the ensemble procedures.

The habitat maps derived from the ensemble approaches (majority and weighted vote) showed an overall accuracy of 94% for the 2003 data (Kappa index of 0.93) and of 95% for the 2016 data (Kappa index of 0.94) (Fig. 2) with no statistical significance of the difference between them ($p > 0.05$). Change analysis was therefore performed using the two ensemble methods.
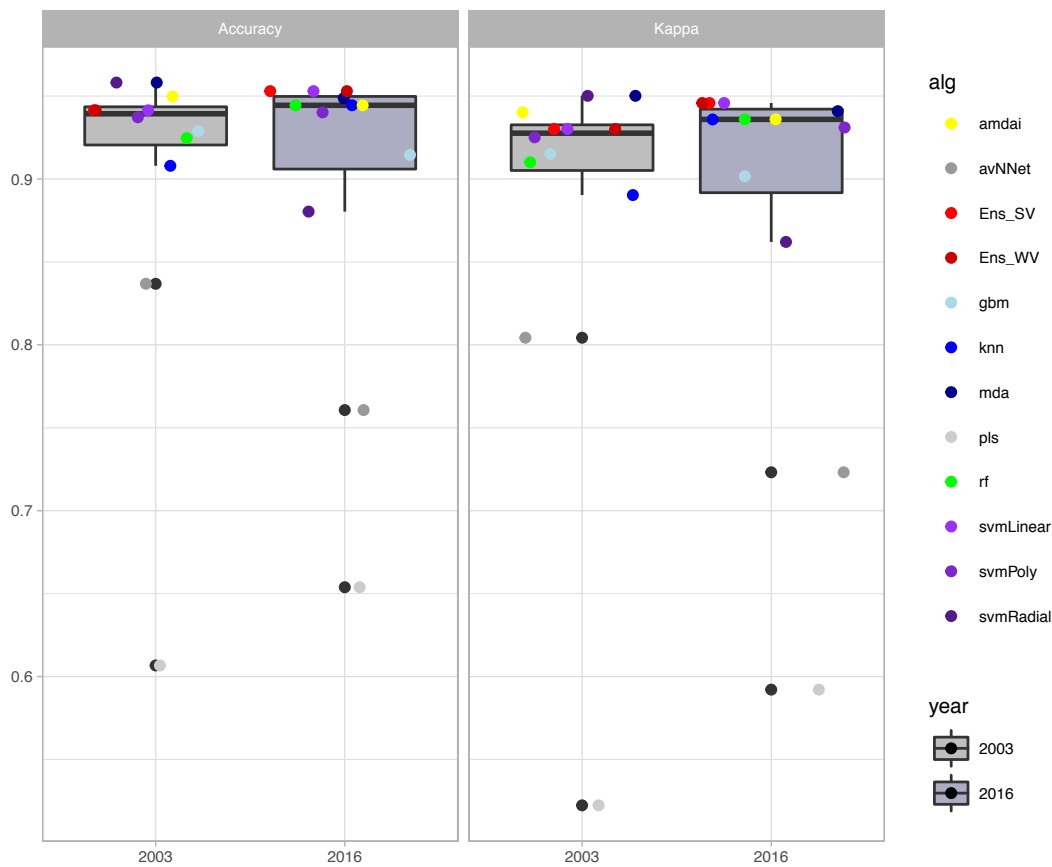

**Figure 2** **Accuracy of habitat maps (overall accuracy and Kappa coefficient) per year and classification method.** amdai (Adaptive Mixture Discriminant Analysis), avNNet (Model Averaged Neural Network), gbm (Stochastic Gradient Boosting), knn (k-Nearest Neighbours), mda (Mixture Discriminant Analysis), pls (Partial Least Squares), rf (Random Forest), svmPoly (Support Vector Machines with Polynomial Kernel), svmRadial (Support Vector Machines with Radial Basis Function Kernel), svmLinear (Support Vector Machines with Linear Kernel), simply voting ('Ens_SV') and weighted voting ('Ens_WV') ensemble approaches. The boxplots display the median, the 50% (box) and 95% (whiskers) confidence intervals.

## Change analysis

The change analysis revealed important temporal dynamics between 2003 and 2016 for the habitat classes identified in the study area (Fig. 5, Table 3). However, the changes depended on the classification algorithm used (Fig. 5). For example, values for water-dependent habitat classes ranged from around 60 ha with most of the classification algorithms, to almost 7 times this value with the 'svmRadial' classifier, clearly indicating overestimation of this unit (Figs. 5–7). The coverage estimated for habitat class dominated by salt marshes and meadows in 2016 ranged from values close to 52 ha with the 'svmRadial' classifier to more than 260 ha with the 'gbm' classifier (Fig. 5).

The habitat maps obtained using the two ensemble classification approaches show a reduction in habitat classes dominated by salt marshes and meadows (24.6–26.5%), natural and semi-natural grasslands (25.9–26.5%) or sand dunes (20.7–20.9%) and an increase in forest (31–34%) and reed bed (60.7–67.2%) in the study area (Fig. 5). In particular, the

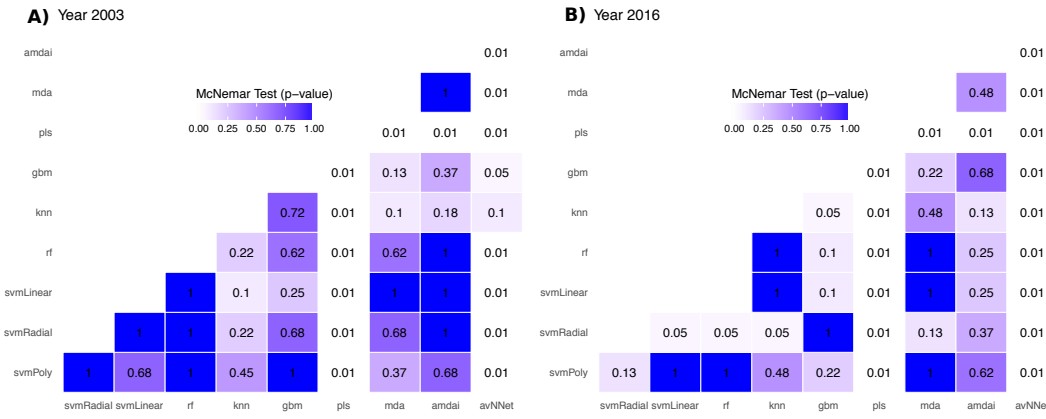

**Figure 3** *P-value from McNemar's tests for each pair of classification algorithm.* amdai (Adaptive Mixture Discriminant Analysis), avNNet (Model Averaged Neural Network), gbm (Stochastic Gradient Boosting), knn (k-Nearest Neighbours), mda (Mixture Discriminant Analysis), pls (Partial Least Squares), rf (Random Forest), svmPoly (Support Vector Machines with Polynomial Kernel), svmRadial (Support Vector Machines with Radial Basis Function Kernel), svmLinear (Support Vector Machines with Linear Kernel). White colours indicate *p*-values lower than 0.01, blue colour intensity increases with the *p*-value.

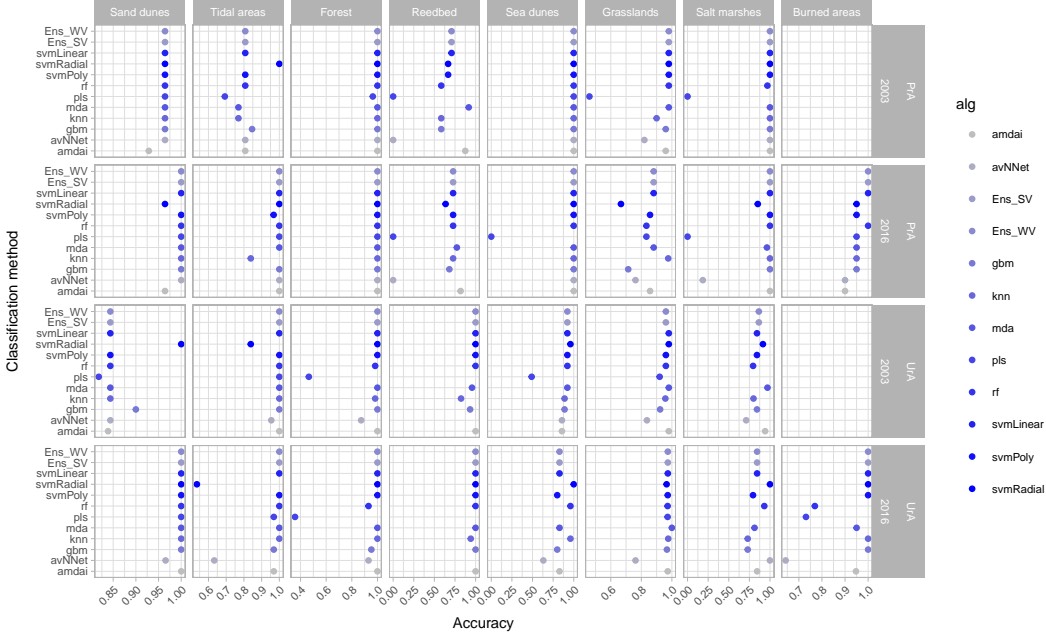

**Figure 4** **Producer's (PrA) and user's (UrA) accuracy per year, habitat class and classification method.** amdai (Adaptive Mixture Discriminant Analysis), avNNet (Model Averaged Neural Network), gbm (Stochastic Gradient Boosting), knn (k-Nearest Neighbours), mda (Mixture Discriminant Analysis), pls (Partial Least Squares), rf (Random Forest), svmPoly (Support Vector Machines with Polynomial Kernel), svmRadial (Support Vector Machines with Radial Basis Function Kernel), svmLinear (Support Vector Machines with Linear Kernel), simply voting ('Ens_SV') and weighted voting ('Ens_WV') ensemble approaches. See Table 1 for habitat classes.

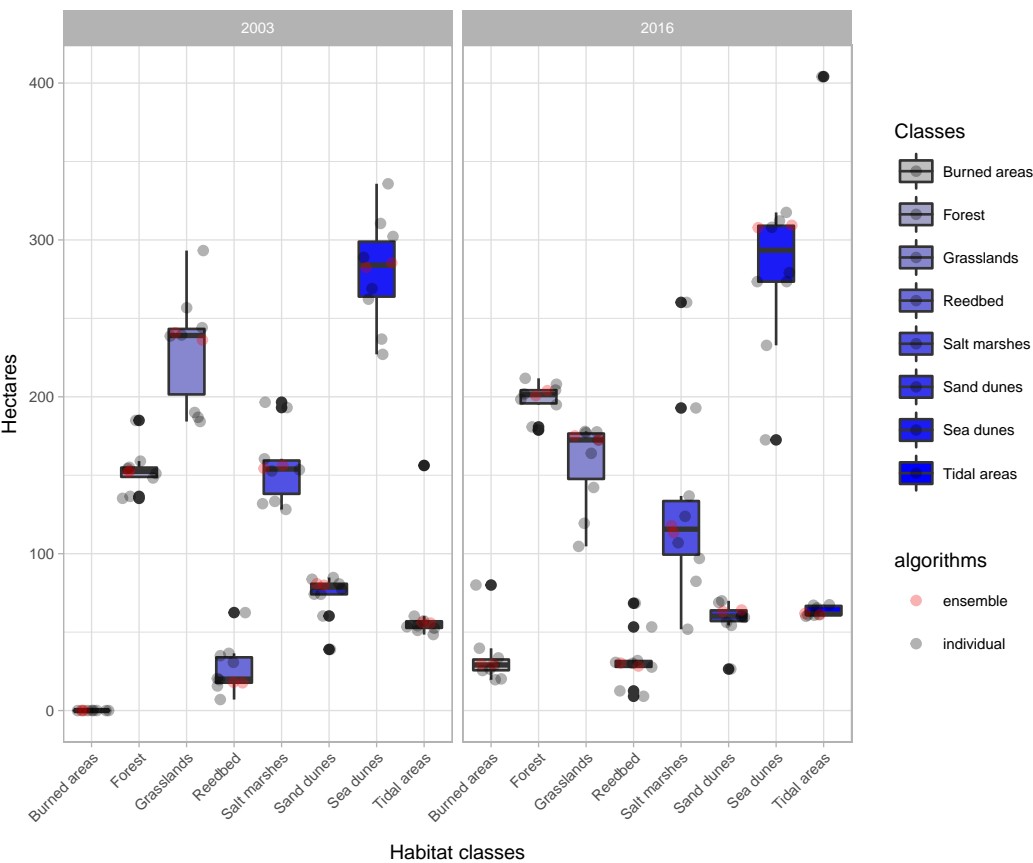

**Figure 5** **Extent (in ha) of each habitat class per year and classification algorithm ('alg').** The boxplots display the median, the 50% (box) and 95% (whiskers) confidence intervals. See acronyms in Fig. 2.

spatial extent of habitat classes characterized by salt marshes and meadows or natural and semi-natural grasslands have largely changed in favour of those dominated by reed bed and forest (Table 3). Burned areas were only identified in 2016, mainly affecting habitat classes with forest, salt marshes and meadows and, to a lesser extent, natural and semi-natural grasslands (Table 3). Areas dominated by 'sea dunes' remained almost unchanged (Fig. 4, Table 3).

## DISCUSSION

Our results confirm the useful role that EO technologies may have in the reporting obligations posed on the Member States by the European Habitats Directive, as well as in the cost-effective monitoring of natural habitats included in the Annex I. This should provide additional support to local managers and decision-makers in relation to the implementation of medium- and long-term conservation measures. However, the uncertainty arising from the large variety of classification methods used may prevent local managers from basing their decisions on EO data. Our results shed light on how different classification algorithms may provide very different quantitative estimates, especially for

**Table 3  Transition matrices obtained from the simply voting ('Ens_SV') and weighed voting ensemble ('Ens_WV') procedures from 2003 (rows) to 2016 (columns) (expressed in hectares) for the study area.** Habitat classes description can be found in Table 1.

|  | Sand dunes | Burned areas | Tidal areas | Forest | Reedbed | Sea dunes | Grasslands | Salt marshes |
|---|---|---|---|---|---|---|---|---|
| Ens_SV |  |  |  |  |  |  |  |  |
| Sand dunes | 53.64 | 0 | 8.01 | 0 | 0 | 18.27 | 0 | 0 |
| Burned areas | 0 | 0 | 0 | 0 | 0 | 0 | 0 | 0 |
| Tidal areas | 2.97 | 0 | 50.4 | 0 | 0 | 0.72 | 0 | 2.07 |
| Forest | 0 | 9.45 | 0 | 128.7 | 1.08 | 0.9 | 9.81 | 2.97 |
| Reedbed | 0.27 | 0.36 | 0 | 1.62 | 7.2 | 2.52 | 4.05 | 1.62 |
| Sea dunes | 6.3 | 4.23 | 2.16 | 4.14 | 2.7 | 249.03 | 7.56 | 9.27 |
| Grasslands | 0 | 6.93 | 0 | 38.43 | 9.36 | 30.78 | 147.51 | 3.15 |
| Salt marshes | 0 | 8.55 | 1.35 | 27.9 | 8.01 | 5.58 | 6.12 | 98.64 |
| Ens_WV |  |  |  |  |  |  |  |  |
| Sand dunes | 54.54 | 0 | 7.92 | 0 | 0 | 18.36 | 0 | 0 |
| Burned areas | 0 | 0 | 0 | 0 | 0 | 0 | 0 | 0 |
| Tidal areas | 2.97 | 0 | 50.31 | 0 | 0 | 0.63 | 0 | 1.89 |
| Forest | 0 | 9.36 | 0 | 128.79 | 1.08 | 0.81 | 9.09 | 2.88 |
| Reedbed | 0.18 | 0.36 | 0 | 2.07 | 7.38 | 2.61 | 4.32 | 1.17 |
| Sea dunes | 6.39 | 4.41 | 1.53 | 3.96 | 3.06 | 248.76 | 6.57 | 7.74 |
| Grasslands | 0 | 6.93 | 0 | 40.5 | 10.71 | 32.31 | 147.69 | 2.7 |
| Salt marshes | 0 | 8.37 | 1.44 | 28.62 | 8.01 | 5.85 | 4.95 | 97.11 |

water-dependent habitats (Fig. 5). In this respect, our findings confirm the need to deal with this uncertainty by using ensemble classification approaches (Figs. 5–7), to effectively improve the accuracy and stability of remote sensing image classification (for a review, see *Du et al., 2012*). Despite these advantages, detailed habitat mapping may require advanced EO technologies (e.g., hyperspatial, hyperspectral, LiDAR) to overcome several constraints that limit the contribution of remote sensing to the reporting obligations of Habitats Directive, such as the spectral similarity of the land covers that belong to different habitat types, the spectral difference of the covers that belong to the same habitat type (*Delalieux et al., 2010*; *Borre Vanden et al., 2011*) or their highly dynamic nature (*Gallant, 2015*).

Our findings showed important changes in the habitat classes over the last 13 years (Fig. 5, Table 3), with potential impacts on natural habitats included in the Habitats Directive (Table 1). The habitat class dominated by salt marshes and meadows has decreased by as much as 25% since designation of the site as part of the Natura 2000 network in 2003. In particular, salt marshes and meadows habitats include *Salicornia* species and other annuals colonizing mud and sand (habitat code 1310), Atlantic salt meadows (*Glauco-Puccinellietalia maritimae*) (habitat code 1330) and Mediterranean and thermo-Atlantic halophilous scrubs (*Sarcocornetea fruticosi*) (habitat code 1420). This habitat class has been negatively affected by wildfires, forest expansion and, to a lesser extent, conversion of the land to natural and semi-natural grasslands (see transitions in Table 3). These patterns can be explained by the concomitant effects of abandonment of traditional agropastoral practices, which may indirectly promote forest spread and expansion (*Stellmes et al., 2013*; *Regos et al., 2015*), a high-frequency fire regime (*Chas-Amil, 2007*) and land-use changes

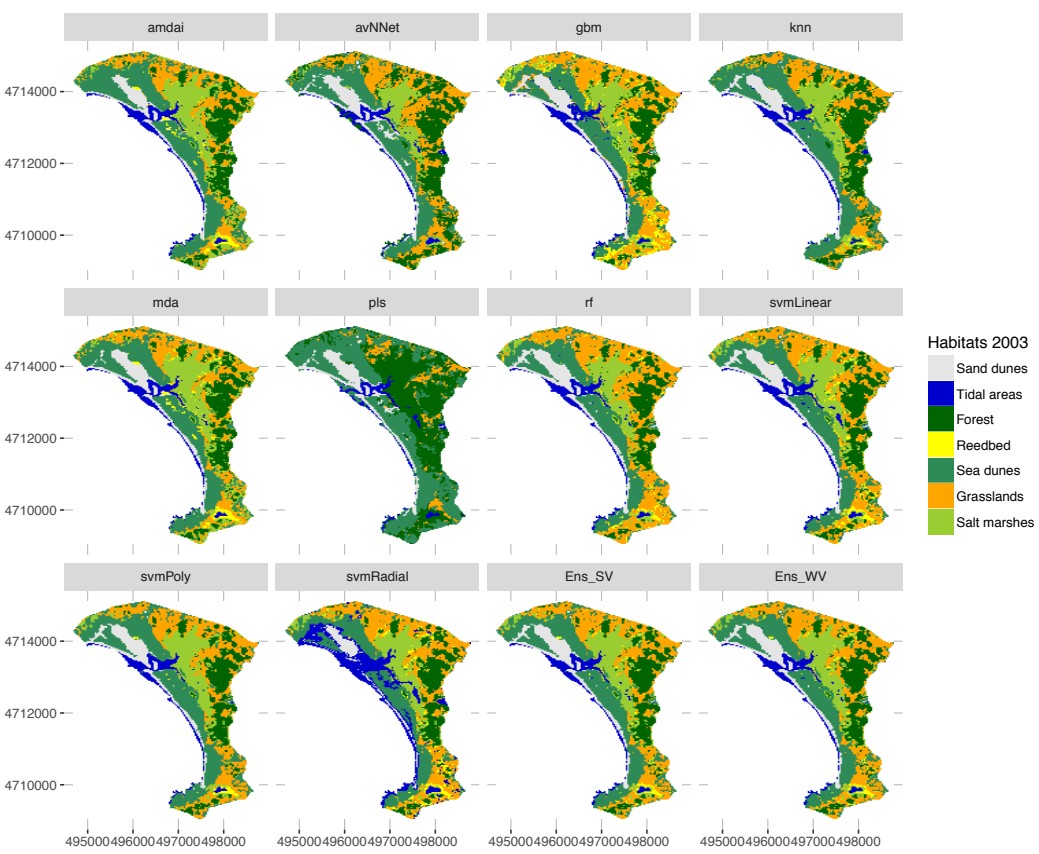

**Figure 6 Habitat maps for 2003 obtained from each classification method.** amdai (Adaptive Mixture Discriminant Analysis), avNNet (Model Averaged Neural Network), gbm (Stochastic Gradient Boosting), knn (k-Nearest Neighbours), mda (Mixture Discriminant Analysis), pls (Partial Least Squares), rf (Random Forest), svmPoly (Support Vector Machines with Polynomial Kernel), svmRadial (Support Vector Machines with Radial Basis Function Kernel), svmLinear (Support Vector Machines with Linear Kernel), simply voting ('Ens_SV') and weighted voting ('Ens_WV') ensemble approaches.

caused by agricultural conversion, as reported for other protected wetlands in southern Spain (*Zorrilla-Miras et al., 2014*).

Natural and semi-natural grasslands have undergone the greatest decline (of up to 26%) since designation of the site as part of the Natura 2000 network. This habitat class includes *Molinia* meadows (habitat code 6410), humid grasslands of the *Molinio-Holoschoenion* (habitat code 6420), hydrophilous tall herb fringe communities (habitat code 6430) and hay meadows (*Alopecurus pratensis*, *Sanguisorba officinalis*) (habitat code 6510). Natural and semi-natural grasslands were mainly replaced by coniferous forest and to a lesser extent by 'sea dune' habitats (Table 3), also indicating afforestation as a main threat. These grasslands were also slightly affected by wildfire in 2016. However, the loss and degradation of these habitats was also related to the gradual expansion of invasive species (*Gonzalez-Martínez, 2014*; *González-Martínez, 2017*). In this respect, new advances in remote sensing technologies and the availability of new sensors with higher temporal, spectral and spatial resolution such as Sentinel-2 from the European Space Agency (ESA)

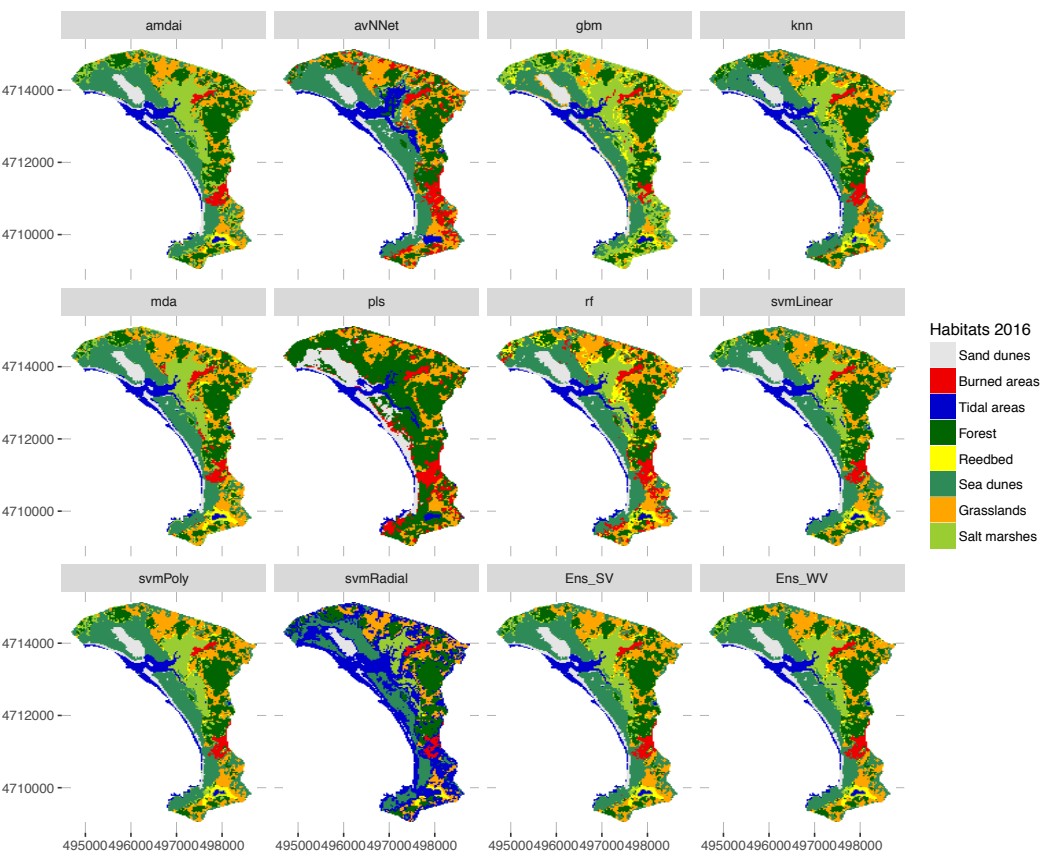

**Figure 7** **Habitat maps for 2016 obtained from each classification method.** amdai (Adaptive Mixture Discriminant Analysis), avNNet (Model Averaged Neural Network), gbm (Stochastic Gradient Boosting), knn (k-Nearest Neighbours), mda (Mixture Discriminant Analysis), pls (Partial Least Squares), rf (Random Forest), svmPoly (Support Vector Machines with Polynomial Kernel), svmRadial (Support Vector Machines with Radial Basis Function Kernel), svmLinear (Support Vector Machines with Linear Kernel), simply voting ('Ens_SV') and weighted voting ('Ens_WV') ensemble approaches.

and low-cost unmanned aerial vehicles (UAVs, also known as drones) should contribute greatly to monitoring of invasive species (*Lehmann et al., 2017*; *Ng et al., 2017*), and to better delineation and mapping of the habitats under EU protection (*Adam, Mutanga & Rugege, 2010*; *Marcaccio, Markle & Chow-Fraser, 2015*; *Stratoulias et al., 2015*; *Gonçalves et al., 2016*).

The area covered by habitat classes dominated by sandbanks, sandflats and shifting dunes ('white and mobile dunes') (see habitat codes in Table 1) decreased by more than 20% relative to the cover in 2003 (Fig. 5), in favour of vegetated dunes ('sea dunes') (Table 3). Sandbanks, mud flats and sandflats are strongly affected by coastal dynamics in the long term and by intertidal fluctuations in the short term. Such dynamics may also have contributed to horizontal displacement of the main 'white dune' (66.5 m in the last decade, see Appendix S1). Despite the losses and gains estimated for 'sea dunes' over the last 13 years, this habitat class (which includes three priority habitats, see description in Table 1) was found to be the most stable over time (Fig. 5, Table 3). This stability has also

important conservation implications for other species listed in the European Directives. For instance, 'grey dune' is the breeding habitat for the Eurasian Stone Curlew (*Burhinus oedicnemus*) (*Domínguez, Otero & Vidal, 2006*), which is included in Annex I of the Birds Directive, and in the Galician Catalogue of Threatened Species.

Reed bed, mainly represented by *Phragmites australis*, has increased greatly since designation of the site as part of the Natura 2000 network (Fig. 5, Table 3). This increase may have been directly favoured by protection of the site and the gradual decline in the traditional reed management (harvesting and burning) by local communities (*Valkama, Lyytinen & Koricheva, 2008*). Although *Phragmites australis* reed bed is not listed in the Habitats Directive, the plant is included in several habitat types of the Annex I, such as estuaries (habitat code 1130), coastal lagoons (habitat code 1150) and inland salt marsh (habitat code 1340) (*Romão, 1996*). Moreover, changes affecting reed bed may also have subsequent effects on other species associated with these habitats (*Valkama, Lyytinen & Koricheva, 2008*), such as the Reed Bunting (*Emberiza schoeniclus lusitanica*) (*Kvist et al., 2011*; *BirdLife International, 2017a*) and the Common Little Bittern (*Ixobrychus minutus*) (*BirdLife International, 2017b*), both included in the Annex I of the Birds Directive and the Spanish and Galician Catalogue of Threatened Species. Therefore, its management and conservation may also have positive effects on species protected by the Birds Directive.

## CONCLUSIONS

Earth observation (EO) technologies may provide cost-effective means of medium- and long-term monitoring of wetland habitats. The proposed methodology is useful for relatively inaccessible sites (e.g., coastal lagoons or reed beds) as it only requires ecological rules based on expert knowledge. Habitat changes can be detected by comparing pairs of maps, and trends can be quantified. This study therefore illustrates how EO-based approaches might be particularly useful to help (1) managers to monitor their decisions in relation to conservation, (2) Member States to comply with the requirements of the European Habitats Directive (92/43/EEC), and (3) the European Commission to monitor the conservation status of natural habitat types of community interest included in Annex I of the Habitats Directive.

### Funding
Adrián Regos was funded by the Xunta de Galicia (post-doctoral fellowship ED481B2016/084-0). The funders had no role in study design, data collection and analysis, decision to publish, or preparation of the manuscript.

### Grant Disclosures
The following grant information was disclosed by the authors:
Xunta de Galicia: ED481B2016/084-0.

### Competing Interests
The authors declare there are no competing interests.

## Author Contributions

- Adrián Regos conceived and designed the experiments, performed the experiments, analyzed the data, prepared figures and/or tables, authored or reviewed drafts of the paper, approved the final draft.
- Jesús Domínguez conceived and designed the experiments, contributed reagents/-materials/analysis tools, authored or reviewed drafts of the paper, approved the final draft.

## Data Availability

The R code is avaliable at http://rpubs.com/ARegos/359655, and the raw data are included in Dataset S1.

## Supplemental Information

Supplemental information for this article can be found online at http://dx.doi.org/10.7717/peerj.4540#supplemental-information.

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
