# Peer review of "The contribution of Earth observation technologies to the reporting obligations of the Habitats Directive and Natura 2000 network in a protected wetland"

_PeerJ, doi:10.7717/peerj.4540_

## Round 0.1 · original submission · Major Revisions

· Academic Editor

Major Revisions

Dear Adrian. Both reviewers see merit and relevance in your work, and I agree. However, some major methodological and reporting issues are flagged. Carefully consider these critiques when revising your manuscript. Adjust your methods as suggested or include sound justification and reflection of the implications of your methodological choices.

Reviewer 1 ·

Basic reporting

The hypothesis that land abandonment causes change in vegetation structure was not properly tested.
Classification maps should be presented. Figure 1 and 4 need improvement, see general comments.

Experimental design

The classification scheme proposed in this study seems not standard and transferable for other sites. The sampling strategy was not mentioned and sample size was not sufficient enough to derive a reliable result. Compasion among different classification algorithms need further statistical test. See general comments for details.

Validity of the findings

The authors drew conclusion that EO-based approach could be used to monitor the conservation status of natural habitat types defined by European Union Habitat Directive, which was not supported by this work. The classification in this study only focused on several groups of land cover classes, while mapping detailed habitat types can be very challenging with conventional remote sensing image classification methods due to their spectral similarities.

Additional comments

Dear authors,

The work presents an Earth observation method to address relevant aspects of reporting obligation towards the Habitats Directive and Natura 2000 network regarding wetland ecosystems. Specifically, the performance of 10 classifiers are compared, the uncertainties in the classification output from these algorithms are explored and an ensemble approach of multiple classifiers is proposed. The case study monitors habitat changes in a single site at local scale by post-classification comparison. I find the topic very relevant, and the manuscript is clearly written in professional, unambiguous language. However, the study shows several limitations that need to be addressed:

• The classification scheme proposed in this study consists of eight classes, and some are grouped categories of detailed habitats defined by European Union Habitat Directive. The definition of the classes (so-called ‘environment unit’ in the manuscript) seems not standard, and it may not be applicable for other sites of Natura 2000 network and hence is difficult to evolve from site-specific research into operational application. I would suggest the authors to adopt a more standard land cover classification scheme and wetland typology (e.g., Ramsar Convention).
• The sample sizes used for training and testing classification are small, especially for reedbed class, which affects the reliability of the high mapping accuracy. In principle at least 30-50 samples should be used for either training or validation. Moreover, I missed the sampling strategy in the manuscript, which needs to be clarified. As Table 2 shows, equal number of samples for each class were used for validation, which may not be representative of the population and causes sample bias. I suggest the authors to adopt proportionate stratified random sampling, and make sure sufficient samples are selected for each class.
• Regarding accuracy assessment, it is also useful to show producer’s and user’s accuracy for each class, especially when you made an inference of overestimation in a certain class (line 248). To compare the performance among different classifiers, differences in overall accuracy is not enough to indicate the superiority. Additional statistical test, such as McNemar’s test (Foody 2004; De Leeuw et al. 2006), is needed to assess the significance of the differences in accuracy. It also provides evidence for the selection of algorithms for ensemble rather than using some arbitrary threshold of 85%.
• For the ensemble approach, majority vote considers different classifiers with equal weights to make the decision regardless of their respective abilities to classify properly. As Du et al. (2012) recommended, weighted vote that uses specific accuracy obtained by individual classifiers as weights would be more suitable in this study.
• It is good that the authors tried to explain some ecological and socio-economic phenomenon related to land cover change. However, the hypothesis that land abandonment causes change in vegetation structure was not properly tested. Plausible explanation of abandonment of traditional agropastroral practices in discussion (line 284) is speculation and inference. Current analysis could not support the hypothesis.
• As a study of land cover classification, it is unprofessional that not a single classification map was included in the manuscript. Besides, it is not clear where the classification was performed even with Figure 1.

References:
Foody GM (2004) Thematic map comparison: Evaluating the statistical significance of differences in classification accuracy. Photogramm Eng Rem S 70: 627–633.
De Leeuw J, Jia H, Yang L, Liu X, Schmidt K, et al. (2006) Comparing accuracy assessments to infer superiority of image classification methods. Int J Remote Sens 27: 223–232.
Du P., Xia J., Zhang W., Tan K., Liu Y., Liu S. 2012. Multiple classifier system for remote
379 sensing image classification: A review. Sensors 12:4764–4792.

Detailed comments:
Line 76: “RAMSAR” should be “Ramsar”-named after Ramsar City in Iran
Lines 196-197, the authors mentioned the grouping of natural habitats into environmental unit based on some expert criteria, which needs further clarification. The reference is in Spanish that can be difficult to understand for general readers.
Line 210: what are lower and upper accuracy values?
Line 248: Could it be other algorithms underestimate this class? With producer’s and user’s accuracy presented, it is more clear to see which class is overestimated/underestimated.
Line 291: “Nature” should be “Natura”
Figure 1: The format of lines needs to be changed for a more clear illustration of the study area. The blue arrow line is not friendly, especially for the west part, and the areas outlined by the blue arrow were not displayed completely.
Figure 4: It is not necessary to use circular plot to show conversion among land covers. It is complex and difficult to estimate the numbers, and similar colors used makes it even difficult to read. Transition matrix can directly and clearly shows the information you need to present.

Reviewer 2 ·

Basic reporting

This is a well done research article which describes an ensemble mapping approach based on satellite images to assess habitat and land cover changes in a Natura 2000 site. This research is indeed quite relevant to help EU member states with their reporting obligations under the EU nature directives. The paper is well written in clear and concise style. Although it could be easily accepted in its present form, I still have a few suggestions which could improve the paper.

Experimental design

No comments

Validity of the findings

No comments.

Additional comments

Line 50 and also the discussion. I agree that your approach could be helpful for the three objectives mentioned but the sentence suggests you actually demonstrated the usefulness of the methodology to provide policy support ("our approaches were found to be useful to help …"). However you did not investigate the usefulness of your approach by surveying potential users so I would slightly reformulate this conclusion in the abstract and also in the manuscript.

The COPERNICUS land product has a high resolution land cover land use data set which has been specifically developed for Natura 2000 sites. However, not all Natura 2000 sites are covered at this point in time. It may be worth checking if data for the site under study is available as an alternative way to validate the outcomes of this study. Or the authors could refer to it in the article. https://land.copernicus.eu/local/natura/natura-2000-2012/view

Figures 2 and 3 are not particularly helpful. I would convert them to tables. In addition I would replace figures 2 and 3 with boxplots with the average and standard deviation calculated over the different EO methods. Also Figure 4 is quite different to read (even if it is nicely designed). Again, habitat conversion and the resulting land flows from one type to another may be best presented in a matrix.

Minor comments:
Line 331: In the EU species are only protected under the Birds and Habitats directives. I have no knowledge of other directives which protect species. For some species, such as Anguilla Anguilla, specific regulations may exist.
Throughout the text: "State Members" should read "Member States" and "benefits and services" should read "services and benefits" (following the order of the ecosystem services cascade model).

---

## Round 0.2 · Minor Revisions

· Academic Editor

Minor Revisions

Some minor remaining issues were flagged by the reviewer, please address & correct.

Reviewer 1 ·

Basic reporting

The manuscript has been much improved since last revision. Some figures and tables need further improvement.

Experimental design

No comment

Validity of the findings

The challenges and limitation in mapping Natura 2000 habitat using remote sensing should be further highlighted in the discussion.

Additional comments

The authors did a good job reworking the manuscript. I only have a few additional minor comments.

1. The authors adopted a more standard classification scheme and wetland typology based on the habitat types defined in the Annex 1 of the directive. I was wondering why reed red was considered as a separate class from salt marshes and meadows. Is the habitat defined in the report by Ramil-Rego et al. 2008? Additionally, in line 342, the authors mentioned that Phragmites australis reed bed was not listed in the habitat directive. But in the Interpretation Manual of European Union Habitats, the plant is included in several habitat types, such as 1130 estuaries, 1150 coastal lagoons and 1340 inland salt marsh.

2. There are several challenges in mapping these Natura 2000 habitats using remote sensing, such as the spectral difference of the covers belong to the same habitat type and the spectral similarity of the covers belong to different habitat types. Detailed habitat mapping may require hyperspectral remote sensing or LiDAR remote sensing. These limitations should be further highlighted in the discussion.
Reference: Borre, J. V., Haest, B., Lang, S., Spanhove, T., Förster, M., & Sifakis, N. I. (2011, 15-17 Sept. 2011). Towards a wider uptake of remote sensing in Natura 2000 monitoring: Streamlining remote sensing products with users' needs and expectations. Paper presented at the 2011 2nd International Conference on Space Technology.

3. Figure 4 caption: user’s accuracy (UsA), but in the figure: UrA

4. Figure 5: since the authors did not intend to differentiate between individual algorithms and between ensemble algorithms, it is better group the legend accordingly (e.g., individual algorithms and ensemble algorithms), otherwise it is a bit confusing showing different items with the same color.

5. Figure 6 and 7 map legend needs improvement, the continuous bar makes it difficult to relate one color to the corresponding class.

6. Dataset S1: EnvUnit should be updated to new habitat class.

---

## Round 0.3 · accepted · Accept

· Academic Editor

Accept

Adequate and prompt revisions, no further comments.